# Defensive denoising methods against adversarial attack

**Sungyoon Lee**
Department of Mathematics
Seoul National University
Seoul, Republic of Korea
goman1934@snu.ac.kr

**Jaewook Lee**
Department of Industrial Engineering
Seoul National University
Seoul, Republic of Korea
jaewook@snu.ac.kr

## Abstract

Deep neural networks are highly vulnerable to adversarial examples. An adversarial example is an image with small perturbation designed to make the networks missclassify it. In this paper, we propose two defensive methods. First, we use denoising methods using ROF model and NL-means model before classification to remove adversarial noise. Second, we perturb images in certain directions to escape from the adversarial area. Experiments on the universal adversarial perturbations(UAP) show that proposed methods can remove adversarial noise and perform better classification.

## 1 Introduction

Recently, deep neural networks show powerful results in the image classification tasks(Simonyan & Zisserman (2014), Szegedy et al. (2015), and He et al. (2016)). However, Szegedy et al. (2013) has shown that small noises that are hard to detect with human eyes can fool the networks.

There are several methods such as FGS(Goodfellow et al. (2015)), IFGS(Kurakin et al. (2016)), and UAP(Moosavi-Dezfooli et al. (2016b)) to generate adversarial examples.

The key idea of the UAP is that the noise with 'directional information' can fool the classifier rather than random noise. Surprisingly, this effect does not depend on a particular image or deep network model. The algorithm involves iterative use of the DeepFool(Moosavi-Dezfooli et al. (2016a)) and projection. DeepFool algorithm is based on that the noise pushing data points near to the decision boundaries can degrade the performance of the classifier and the magnitude of this noise can be considered to measure the classifier's robustness. The goal of the algorithm is not to find the nearest decision boundary, but to find a sufficiently close boundary. It is therefore safe to assume that boundaries can be approximated as linear hyperplanes.

In this paper, we propose new methods to solve this problem by removing noise of the perturbed images. We implement ROF modelRudin et al. (1992)) and NL-means model(Buades et al. (2005)) to remove such perturbations and show the result of the methods according to denoising degree.

...

## 2 Proposed Methods

In order to defence against attack using adversarial noise, we first consider denoising before classification model. We use two noise removal algorithms(ROF and NL-means). The UAP contains the directional and metric information of decision boundary and this perturbation is not concentrated in the local area of the image. This kind of noise can be easily removed with denoising models. ROF and NL-means models effectively forget this information without changing the classification performance on the undamaged images.

Image can be represented as a vector in the high-dimensional space $v \in \mathbf{R}^N$. Thus, an imperceptible noise, $\delta$ as a vector with elementwise small value(infinite norm sense) can make new image vector further away from the original one in the euclidean space($l^2$ norm sense). Denoising also generates

very different image vector $D(v + \delta)$ from $v + \delta$. However, classifier is rather robust to denoising. Moosavi-Dezfooli et al. (2016b) and this result show that deep neural models are vulnerable to noise in certain directions. The natural question then arises, is there a certain direction that can make classifier perform better? We assume that this 'good' perturbation can be generated by averaging random perturbations that lead classifier to change their prediction correctly when it was classified incorrectly or to predict more confidently when it was classified correctly as in 1. In the equation, $L_y(v)$ is a loss function for input v and target y. In the experiment, we use the random perturbations that satisfies the first case.

$$v_{good} = \frac{\sum_i \mathbf{I}(L_y(v + \delta_i) < L_y(v))\delta_i}{\sum_i \mathbf{I}(L_y(v + \delta_i) < L_y(v))} \quad \text{where} \quad \delta_i \sim N(0, \epsilon I) \tag{1}$$

### 2.1 Denoising methods: ROF and Non-local means

There are several techniques for removing noise from an image. The classical ROF model, also known as 'Total variation denoising', is an algorithm that can smooth the noise by reducing total variation. This noise removal method is widely used because it is simple and fast. However, important features such as texture can often be oversmoothed. This drawback can be solved by a method called non-local means. The non-local means algorithm preserves fine structure using non-local means as in Eq. 2. In the equation, $\Omega$ is the image area, $v$ is the unfiltered image, $u$ is the filtered image, $f$ is the weighting function, and $Z(p)$ is the normalization factor:

$$u(p) = \frac{1}{Z(p)} \int_\Omega v(q)f(p,q)dq, \quad \text{where} \quad Z(p) = \int_\Omega f(p,q)dq \tag{2}$$

## 3 Experiments

We use 10K ImageNet training data(Russakovsky et al. (2015)) to train the UAP, $\delta_{UAP}$ and 10K ImageNet validation data to test the proposed methods. For denoising parameters, we use a weight of 10 for ROF, a patch size of 2, a patch distance of 3, and a cut-off distance of 0.05 for NL-means. As a classifier, we use the pre-trained inception V1(Szegedy et al. (2015)) model as target model and use VGG-16 and VGG-19(Simonyan & Zisserman (2014)) as black box attack models. A typical example for ROF model is shown in Fig. 1.

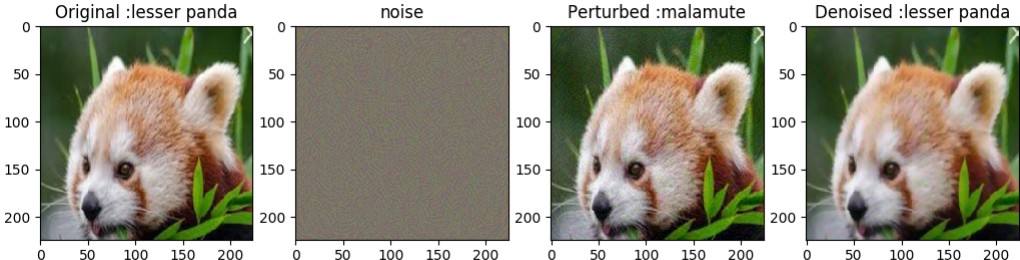

Figure 1: The ROF model can remove the universal adversarial noise and shows correct classification for the most cases(see details in Appendix). The classified label on the top of each image(True = lesser panda). From left to right: Original image, noise(UAP), Perturbed image, Perturbed image after denoised

The ROF model minimizes the total variance $J(u)$ remaining the variance $\sigma(u)$ from $v$ is smaller than given constant $\sigma$, so $u$ is in the $l^2$ ball with radius $\sigma$ centered at $v$. Therefore, the result image is heavily dependent on the value of $\sigma$ we control by denoising weight. We applied different weights(5, 10, 20) and the results are shown in Fig. 2. For NL-means, we control the degree of denoising by changing cut-off distance $h$, which can be considered as a width parameter in Gaussian kernel. Using the NL-means with larger $h$, we get more blurry image.

Table 1: Classification accuracy for the case of clean, perturbed, denoised(ROF, nlm) image, denoised image with additional noise. The target model is IncV1 and each column represents the attack model.

| | IncV1 | VGG16 | VGG19 |
|---|---|---|---|
| $v_0$ | | 69.4 | |
| $v_0 + \delta_{UAP}$ | 12.5 | 28.7 | 31.7 |
| $D_{ROF}(v_0 + \delta_{UAP})$ | 56.3 | 47.2 | 49.1 |
| $D_{ROF}(v_0 + \delta_{UAP} + v_{good})$ | 57.1 | 51.2 | 51.3 |
| $D_{nlm}(v_0 + \delta_{UAP})$ | 57.7 | 45.0 | 46.9 |
| $D_{nlm}(v_0 + \delta_{UAP} + v_{good})$ | 58.0 | 47.7 | 49.4 |

Additional noise, $v_{good}$ is trained using the Eq. 1 and 10K ImageNet validation data that are disjoint from the test set. As shown in the Table 3, this additional noise slightly but always increases classification accuracy.

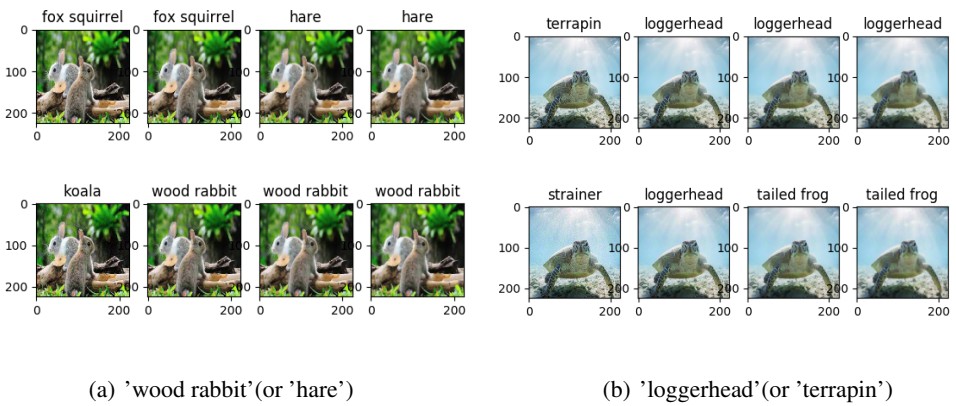

(a) 'wood rabbit'(or 'hare')          (b) 'loggerhead'(or 'terrapin')

Figure 2: The weight parameter can change the classification performance. In the most cases, classifier is robust to the weights within the range we consider(see details in 3). The classified label on the top of each image. Top: Original, Bottom: UAP perturbed image, From left to right: baseline image, $w = 5$, $w = 10$, and $w = 20$

ACKNOWLEDGMENTS

This work was supported by the National Research Foundation of Korea (NRF) grant funded by the Korean government (MEST) (No. 2016R1A2B3014030) and (MSIP)(No.2017R1A5A1015626)

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

APPENDIX

A. DENOISING RESULTS

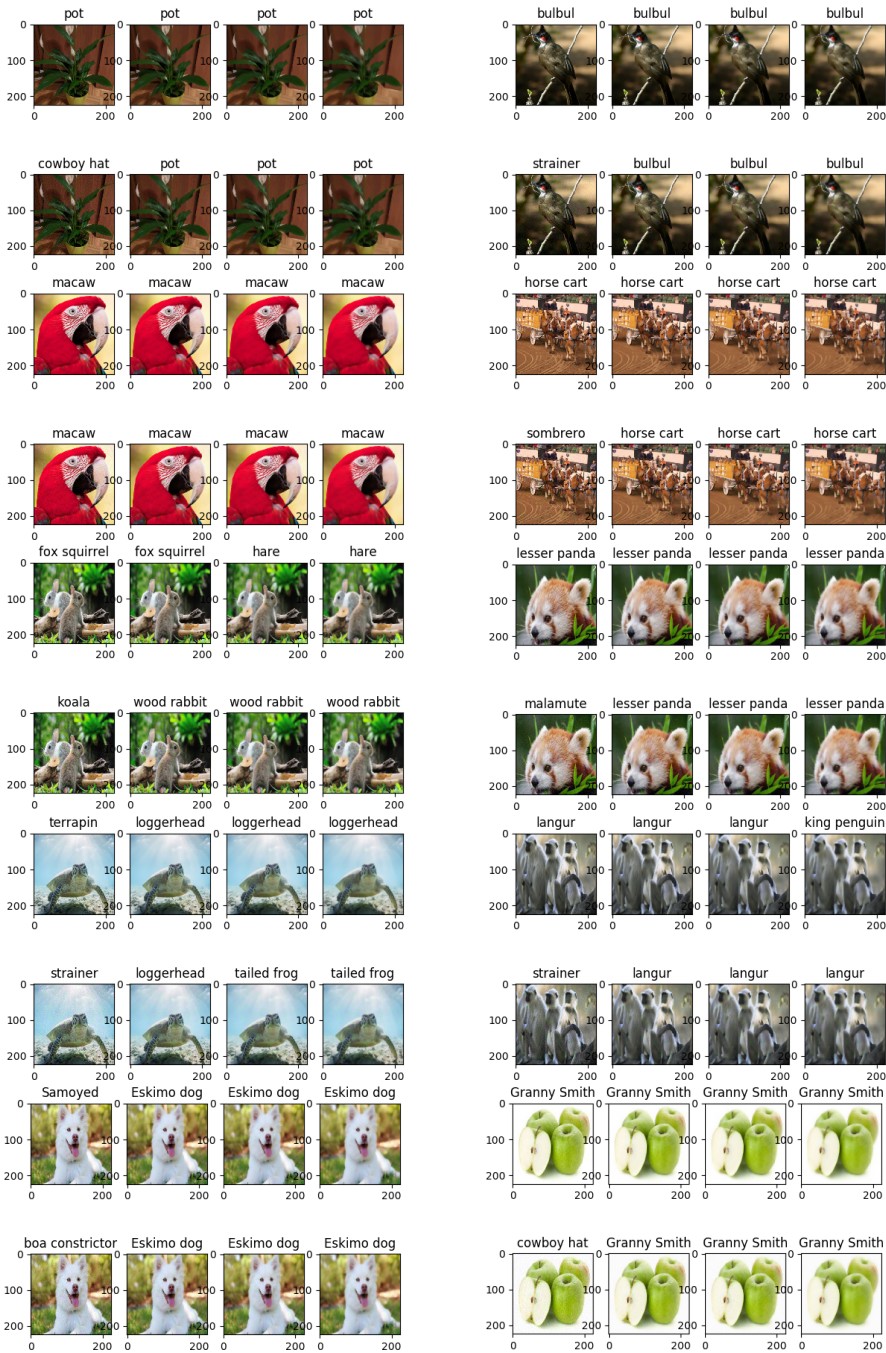

Figure 3: The experiment result for ROF model with the classified label on the top of each image. Top: Original, Bottom: UAP perturbed image, From left to right: baseline image, $w = 5$, $w = 10$, and $w = 20$

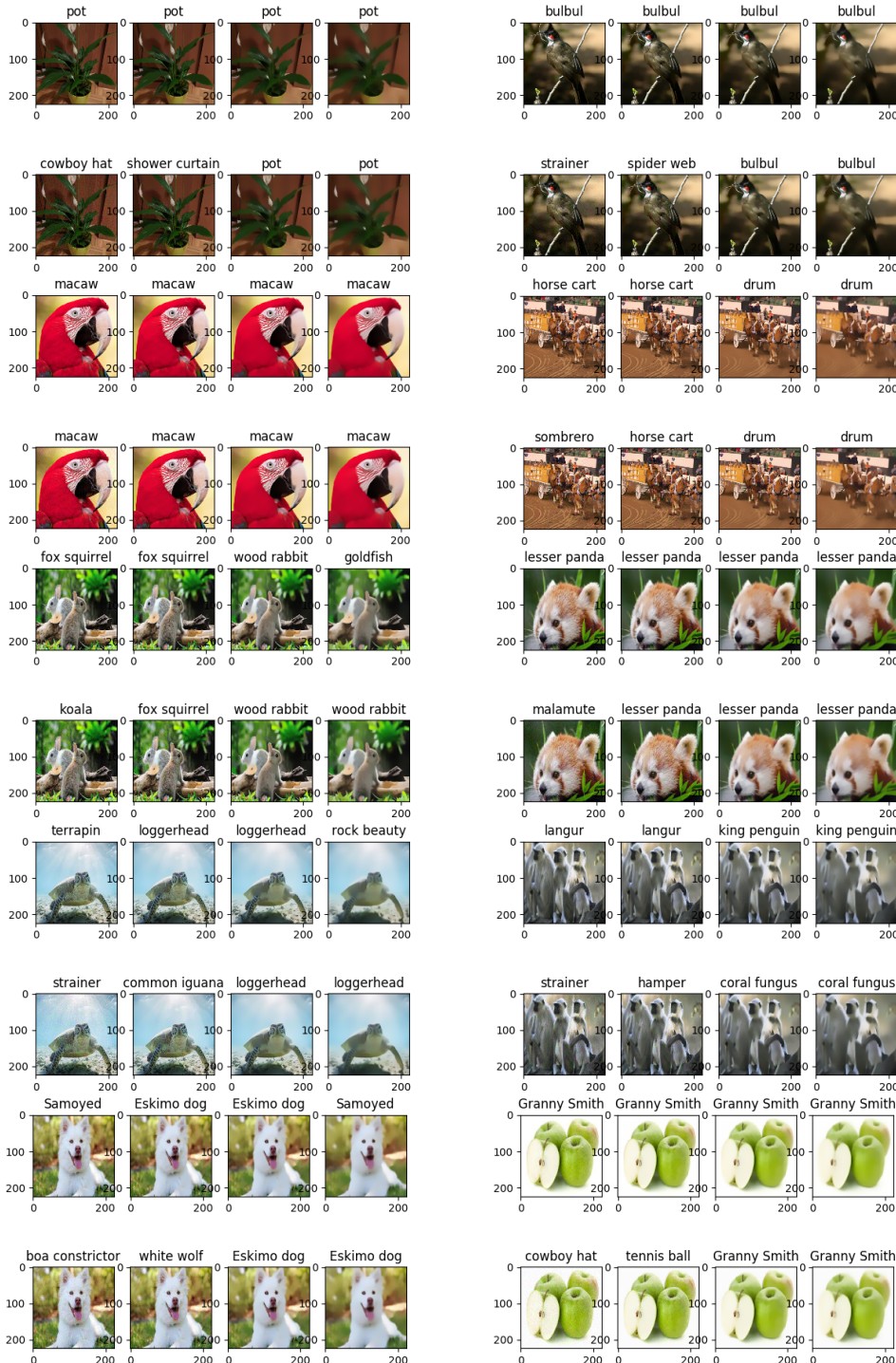

Figure 4: The experiment result for NL-means model with the classified label on the top of each image. Top: Original, Bottom: UAP perturbed image, From left to right: baseline image, $h = 0.02$, $h = 0.05$, and $h = 0.1$

