# OpenReview forum: "Defensive denoising methods against adversarial attack"
_ICLR.cc/2018/Workshop — Reject_

### Official Review · AnonReviewer3 · 2018-03-11

**Rating:** 6
**Confidence:** 4

**Review:**

In this paper, the authors consider the use of denoising methods for defense against adversarial attack.

The attack is obtained by adding noise using universal adversarial perturbations. This is defended against by considering classical denoising methods of ROF TV norm based denoising and NL means based denoising.

The authors also consider the case of adding a small amount of noise that is obtained through training can after denoising be added to further improve the performance.

Thorough comparison would perhaps benefit by considering which denoising methods adversely affect the method. One question is whether any such denoising be valid and considering a systematic evaluation for these. Another question is how sensitive is the method to the selection of parameters. Some analysis on this would also be relevant.

---

### Official Review · AnonReviewer1 · 2018-03-12
**Filtering Adversarial Examples**

**Rating:** 4
**Confidence:** 5

**Review:**

This work proposes a smoothing operation, effectively a low pass filter, on adversarial examples. The experimental evidence are scarce and unconvincing. There is no theoretical justification for why this would work. The manuscript is poorly written.

---

### Decision · Program_Chairs · 2018-03-20
**ICLR 2018 Workshop Acceptance Decision**

**Decision:**

Reject

**Comment:**

Based on the reviews, this paper has not been accepted for presentation at the ICLR workshop. However, the conversation and updates can continue to appear here on OpenReview.